# Oracles in Decentralized Finance: Attack Costs, Profits and Mitigation Measures

**DOI:** 10.3390/e25010060

**Published:** 2022-12-28

**Authors:** Ayana T. Aspembitova, Michael A. Bentley

**Affiliations:** Euler Labs, London EC1V 2NX, UK

**Keywords:** DeFi, oracle, automated market makers, decentralized exchange, lending protocol

## Abstract

Decentralized finance (DeFi) is by far the most popular application of blockchain technology. Despite the wide acceptance of new financial instruments and services, there are still many unexplored areas in the field. We dedicate this research to the understanding of one of the most crucial limitations of decentralized finance—oracles. DeFi protocols, as well as other blockchain applications, function in a closed environment and regularly need to fetch real-world information (e.g., assets’ prices)—the tool used for this purpose is called an oracle. We review the existing oracle types in DeFi applications and focus our research on the least explored one: when another protocol, typically a decentralized exchange, serves as a price oracle. After explaining the mechanisms behind the decentralized exchanges, we introduce an algorithmic model that allows one to safely design a decentralized oracle and adjust crucial parameters. We believe that understanding and implementing the logic presented in the model can help to reduce the chances of price manipulations attacks, which are the most frequent incident types in DeFi.

## 1. Introduction

Blockchain-based smart contracts have been successfully growing, and their use cases are quite innovative and have attracted lots of interest valued in the billions of dollars. However, there is a fundamental limitation of decentralized applications—they execute in a closed environment and a bridge service (oracle) is needed when obtaining information outside of the blockchain. As decentralized applications evolve and mature, oracles play an increasingly prominent role in ensuring the safety across smart contracts. Despite the critical role that oracles play in decentralized applications, the research is still in its infancy. In [1], the authors performed a bibliometric analysis and demonstrated the alarming scarcity of the research dedicated to blockchain oracles. Moreover, in the recent study of DeFi incidents [2], the authors empirically showed that oracle manipulation attacks are the most frequent incident types in DeFi. Although there are tools that can detect the price manipulation attacks [3,4], and identify new vulnerabilities in real time, there is still a need for prevention measures. The lack of understanding of oracles mechanics and functions concerns not only academic research but more so the real users of decentralized applications.

Decentralized finance (DeFi) uses blockchain technology to provide financial instruments without intermediaries in a trustless and transparent manner [5]. DeFi covers a wide range of financial products, offering innovative alternatives to traditional financial products, such as stablecoins, exchanges, lending protocols, insurance and yield farming protocols.

Here, we provide an overview on why DeFi rests heavily on the use of oracles and how information from the outside world can be retrieved. Generally speaking, there is some *ground truth* information that resides outside of smart contracts, and smart contracts need it for the proper performance. To obtain such ground truth, smart contracts need reliable *data sources*—any entity that stores the ground truth information (databases, sensors or other smart contracts). Then, *data feeders* report off-chain data to an on-chain system. The systematic explanation on the existing type of oracles in a blockchain is provided in [6]. As for the decentralized financial applications, the ground truth needed is the price of the assets listed in a smart contract. Although there are many types of oracles with different functions and characteristics, the oracles currently used in DeFi can be broadly divided into two main categories—decentralized trust-based oracles and decentralized exchanges used as oracles.

Decentralized trust-based oracles function as a smart contract and do not rely on a single source of information. Instead, they query multiple sources and aggregate the obtained information into a single output. The papers [7,8,9] provided a detailed review on the architecture, workflow and weak points of various decentralized oracles, such as Chainlink [10,11], Provable [12], Oraclize, etc. Some DeFi applications are fetching the price information directly from the decentralized exchanges by either getting the spot price or aggregating the prices over a certain window size. Using the spot price can be very dangerous because the price can be easily manipulated [13,14,15]. Therefore, more and more DeFi applications started using the TWAP (time-weighted average price) instead—the output price is calculated as a weighted average over a certain time period and, therefore, the cost of price manipulation of the TWAP oracle increases linearly with the length of the TWAP averaging window, reducing the chance of an oracle hack.

In this paper, we focus on the decentralized exchanges (DEXs) used as oracles for DeFi protocols. While trust-based oracles have attracted some attention from the researchers, using DEXs directly as oracles is still not well understood. In [16], the authors analyzed the cost of TWAP manipulation when an arithmetic mean is used for the aggregation and also considered the possibility of an MMEV attack. Decentralized exchanges utilize the concept of automated market makers (to be explained in detail in Section 2). Our main contributions consist of the following: we systematize the existing knowledge about using automated market maker (AMM)-based decentralized exchanges as oracles, we derive attack costs for the most popular cost functions used in DEXs, then we derive the relations between protocol-specific parameters and oracle-specific parameters that impact the safety of using the DEX-based oracle and, finally, we develop the algorithmic model that allows to assess the risks of using oracles in a given protocol. Overall, knowing the mechanics behind the oracles’ work would give a comprehensive understanding on how attacks can be performed. Implementing the logic presented in the model below would give the quantitative estimate on the cost a potential attacker needs for a successful attack. Knowing the mechanism behind the price oracle and being able to precisely estimate the cost of a potential attack provides an additional layer of security to the protocols using DEX-based oracles.

The paper is structured as follows. First, we review the most popular AMM-based decentralized exchanges, demonstrate their logic and the cost functions used for asset pricing. In the appendices, we derive the cost of the attacks for each type of AMM pricing function discussed in Section 2. Then, in Section 3, we discuss various aggregation methods that can be used in DEX-based oracles and show how they can be impacted by the price manipulation attack. Section 4 aggregates all the information obtained above and provides a step-by-step algorithm on how to mitigate attacks related to the DEX-based oracles on the example of a lending protocol. We simulate various attack scenarios to the lending protocol on two types of AMM cost functions—a constant product and stableswap. Finally, we conclude all the findings and discuss the future directions of this research in the last section.

## 2. Automated Market Makers

To understand the safety of using the DEX-based oracle, we need to first be familiar with how DEXs work and understand the mathematics behind it—the cost function utilized by the DEX. In this section, we first explain the mechanism of the decentralized exchange protocol and then review the popular AMM cost functions and demonstrate how they are used to price assets in a DEX.

An AMM-based decentralized exchange consists of pools of different assets (liquidity pools). Liquidity to these pools is provided by people who wish to gain income from the transaction fees (liquidity providers). Each pool can have few assets (currently most of the pools have two assets) and users who want to exchange assets (traders) interact directly with the given pool to swap asset *x* to asset *y*.

In centralized exchanges, the price discovery happens by matching the sell and buy orders from various counterparties. In contrast to it, decentralized exchanges are based on the automated market-making mechanism (AMM). The AMM utilizes the cost function that discovers the price algorithmically—this function only allows counterparties to exchange the assets for the prices along the trajectory determined by the AMM formula and quantities of the available assets. Although the implementation of AMM functions to price assets in decentralized exchanges is quite novel, the idea of agents automatically placing bets and following prescribed rules is not new and has been implemented in many areas to aggregate the information—the prediction of building openings [17], sport matches [18], etc. Overall, the idea of automated market making is to define algorithmic rules for agents within the system to place their bets on a certain subject, aggregate them and derive a single function (*conservation function*) from the outcome. In DEXs, this process goes a little different. First, the conservation function is defined and then agents (traders and liquidity providers) match their trades, and whenever there is a trade that goes beyond the expectations of the cost function, it is punished by the algorithm and, therefore, it discourages agents to behave (trade) differently than prescribed by the conservation function. Although in DEXs agents are not algorithmic bots but real people, they do act in a way as was expected algorithmic bots to act to preserve the cost function.

In [19], Othman introduced five desideratas (desirable properties) for cost functions—monotonicity, convexity, bounded loss, translation invariance and positive homogeneity. As it was proven by [20], it is impossible for the cost function to satisfy all five properties. Therefore, all the cost functions utilized by AMMs satisfy only a few properties, while others are relaxed.

### 2.1. Logarithmic Market Scoring Rule

The first automated market maker for prediction markets was introduced by Hanson [21,22]. It has been quite popular due to its simple analytical form and satisfying the main desirable properties for cost functions (convexity, bounded loss and translation invariance).

The Logarithmic Market Scoring Rule (LMSR) conservation function for *n* assets is defined as:(1)$$C\left(x\right)=b\log(\sum_{i=1}^{n}exp\left(x_{i}/b\right)$$
where b>0 is the liquidity parameter, it is strictly positive, constant and it is defined before the pricing of assets. *b* parameter controls the liquidity in the market—the higher the *b*, the less the price is shifted when assets are added. Moreover, it translates into the bigger maximum loss because the market maker’s worst-case loss is the function of *b* which is *b* log *n*.

The derivative of the cost function C(x) is the price function in the LMSR:(2)$$p_{i}\left(x\right)=\frac{exp(\frac{x_{i}}{b})}{\sum_{j}exp\left(\frac{x_{j}}{b}\right)}$$

The LMSR is used in many settings, such as auctions, prediction markets, rating markets, etc. In decentralized finance, the LMSR has not been widely used for a few reasons: first, the LMSR does not satisfy the liquidity sensitivity property; second, it is quite easy and cheap to compromise the price of an asset when the LMSR is used as a cost function.

By allowing the parameter *b* to be the function of the outstanding quantities instead of being constant, the LMSR becomes liquidity sensitive—the Liquidity-Sensitive Logarithmic Market Scoring Rule (LS-LMSR), introduced by Othman [23]:(3)$$C\left(x\right)=b\left(x\right)\log(\sum_{i=1}^{n}exp\left(x_{i}/b\left(x\right)\right)$$
where function *b* is as follows:(4)$$b\left(x\right)=\alpha \sum_{i}x_{i}$$
where α is the parameter that is strictly positive and set before the pricing of assets. The possible maximum commission (also called vigorish *v*) depends on the α parameter and does not exceed *v* when α is set as follows [24]:(5)$$\alpha =\frac{v}{n\log n}$$
where *n* is the number of outcomes (assets in the pool for AMM-based DEX). Depending on what is the desired maximum commission *v*, the optimal parameter α can be easily calculated.

In decentralized finance, the LS-LMSR is used in applications such as Augur [25] and Gnosis [26].

### 2.2. Constant Product Market

Constant product AMM (CPAMM) cost function for *n* assets is defined as follows:(6)$$C\left(x\right)=\prod_{i=1}^{n}x_{i}$$
where C(x) set as a constant.

Constant product AMM has many advantages that makes it suitable to be used in DEXs—it is simple to code into the smart contract, it is a convex function which meets the principles of supply and demand and it is also liquidity sensitive. Although it has been shown in [27] that prices in such a DEX can be inaccurate during volatile markets, this cost function still remains the most popular and being utilized by large DEXs, such as Uniswap [28,29].

Decentralized exchange pools consist of two tokens and Equation (Equation 6) becomes the following:(7)x×y=k
where *k* is the constant, *x* is the amount of the first token and *y* is the amount of the second token.

The price for each token in a pool can be calculated by simply dividing the number of tokens in one reserve to the number of tokens in another. A more detailed review of constant product markets is given in [30,31].

### 2.3. Combination of Constant Sum and Constant Product Markets

In DeFi, there are assets that have the same value, for example, a different version of USD (USDC, USDT, etc.). Because the ratio between asset *x* and asset *y* in such pools is stable and close to 1, they are called stableswap pools. The pricing formula for stableswap pools was developed by the Curve team [32]. Essentially, this is a combination of the constant product market pricing formula xy=k and the linear invariant x+y=C. The rationale behind adding the linear invariant term to the constant product formula is to achieve the closer peg 1:1 and allow lower slippage for stableswap pools. When using only a linear invariant formula, tokens are always traded at 1:1 with zero slippage; however, this might lead to the depleting of the pool’s one token. Using only a constant product formula leads to larger slippage and a less stable peg. Therefore, the combination of these two curves allows to keep the pool balanced while providing a more stable peg.

The final stableswap curve formula looks as in Equation (Equation 8). For the full explanation and derivation, refer to the paper [33].
(8)$$2^{2}A\left(x+y\right)+D=2^{2}AD+\frac{D^{3}}{2^{2}xy}$$
where *A* is the amplification factor for the linear invariant curve—the larger the *A*, the closer the curve to the linear. *D* is the total amount of tokens in the pool.

To calculate the price for the token, one needs to express the curve for *y* from Equation (Equation 8), and the derivative of that expression stands for the price. Stableswap AMM is widely used in many DEX pools that have the same price for both tokens.

## 3. Aggregation Methods

In this paper, we focus on oracles for DeFi applications that get price information directly from the decentralized exchanges by aggregating the output prices over a certain time period. Every time there is a new swap (trade) in the DEX, the price is updated in oracle and then the time-weighted average is calculated. To mitigate the possible effect of the price manipulation within one or a few blocks, one would prefer to use the time-weighted and/or liquidity-weighted average price.

In this section, we discuss various aggregation methods and show the impact of a price manipulation attack on each of them.

### 3.1. Arithmetic Mean Time-Weighted Average Price

The arithmetic mean TWAP over *n* price updates is calculated as follows:(9)$$TWAP=\frac{\sum_{i=1}^{n}t_{i}p_{i}}{\sum_{i=1}^{n}t_{i}}$$
where $t_{i}$ is the time elapsed between the price update *i* and next price update i+1, and $p_{i}$ is the price during that period. *n* is the averaging window.

We estimate the effect of manipulation on the TWAP price when the attacker wants to consistently manipulate the spot price for *m* times of price updates within the averaging window *n*.
(10)$$TWAP_{m}=\frac{\sum_{i=1}^{n-m}t_{i}p_{i}+\sum_{j=n-m+1}^{n}t_{j}p_{j}}{\sum_{i=1}^{n}t_{i}}$$
assuming that attack would happen in the last *m* blocks. From here, we would like to estimate the $p_{j}$—how big should the manipulated price be that the attacker should target in order to achieve the desired effect on $TWAP_{m}$.
(11)$$\sum_{j=0}^{m}t_{j}p_{j}=TWAP_{m}\times \sum_{i=1}^{n}t_{i}-\sum_{i=1}^{n-m}t_{i}p_{i}$$
In the case when the attacker does not want to be exposed to arbitrageurs and wants to manipulate the price within one block m=1, the manipulated price will be as follows:(12)$$p_{j}=\frac{TWAP_{m}\times \sum_{i=1}^{n}t_{i}-\sum_{i=1}^{n-m}t_{i}p_{i}}{t_{j}}$$
Although using the TWAP instead of a spot price is safer in terms of avoiding the malicious price manipulations, the output from the averaging might not be accurate.

### 3.2. Geometric Mean Time-Weighted Average Price

The geometric mean TWAP over *n* blocks can be calculated as the *n*th root of the product of the spot price on each block:(13)$$TWAP={\left(\prod_{i=1}^{n}p_{i}\right)}^{\frac{1}{n}}$$

If the attacker wants to manipulate the geometric mean TWAP by manipulating the price over *m* blocks, then the target TWAP will be calculated as follows:(14)$$TWAP_{m}={\left(p^{n-m}\times q^{m}\right)}^{\frac{1}{n}}$$

An attacker wanting to manipulate the TWAP to some particular oracle price $TWAP_{m}$ over *m* blocks will need to know what spot price *q* they need to move the normal spot price *p* to in each of those blocks. It can be calculated by rearranging Equation (Equation 14):(15)$$q=\sqrt[{m}]{\frac{TWAP^{n}}{p^{n-m}}}$$

This equation shows that it is surprisingly difficult to move the geometric mean TWAP from the wider market spot price when manipulated blocks are few in number relative to unmanipulated blocks. That is, the spot price must be moved a significant distance from its wider market price in order to have even a modest impact on the geometric mean TWAP.

### 3.3. Median Time-Weighted Average Price

Using median time-weighted average prices as oracles has been discussed in [34], although they have not been widely implemented in practice yet. Theoretically, because the median is unaffected by the effect of outliers, it could be a solution to avoid single-block manipulation attacks, especially in pools with small liquidity. For an attacker to influence the oracle’s final output price, they would need to control the last *m* block prices for at least half of the period of the window size.

From the economic point of view, storing price time series over a certain period to calculate the median could be very expensive in terms of the gas cost in the Ethereum blockchain. In alternative blockchains with a different technical design and cheaper gas cost, this could be possible if the pros of using the median TWAP outweigh the cons. In DeFi, median TWAP oracles have been implemented in the Euler Finance protocol as an alternative price source to the geometric mean TWAP [34].

## 4. Algorithmic Model to Estimate the Safety of TWAP Oracle

We have reviewed AMM cost functions in decentralized exchanges that are often used as price oracles. Moreover, we looked at the most popular methods for price aggregation in oracles. Now, we would like to systematize everything into the algorithmic model that allows to estimate the safety of any DEX-based TWAP oracle. Inputs in the model are parameters of the protocol using TWAP oracle and parameters of AMM that is being used as oracle. Outputs of the model are the attack cost AC and capital *C* needed to provide to the protocol under attack to be able to profit from it. Overall, the model output only tells how much funds a potential attacker needs to gain the profit from price manipulation. To assess the economic feasibility of such an attack, one would need to perform additional independent analysis, for instance, if model output tells that attack would cost 100 USD, then it is economically feasible for many people and, therefore, it is not safe. If model estimates AC and *C* to be big so only a few people can technically perform an attack, then one can conclude that chances of oracle attack are low. Outputs from the model can serve as a starting point to decide on the safety of TWAP-based oracle.

We first introduce the general algorithmic model that allows to estimate the feasibility of price manipulation attack. Then, we explain how the model can be implemented based on the example of lending protocol—this is the most popular use case of TWAP oracles. We provide a brief explanation of how lending protocols work, demonstrate how model can be used and simulate various attack scenarios for constant product and stableswap AMM-based TWAP oracles.

### 4.1. Algorithmic Model

Algorithm 1 shows step by step how to estimate the safety of TWAP oracle. Model outputs are attack cost AC and minimum collateral *C* an attacker needs to profit from their manipulation. Knowing these parameters allows us to estimate the economic feasibility of such an attack and, therefore, to decide whether it is safe to implement the given DEX as an oracle. Input parameters needed for the model can be divided as protocol specific, oracle specific (averaging window size WS) and DEX specific (liquidity *L*). These parameters can be easily found in protocol’s web-page and in DEXs page (*L*). To find values of Δx (number of tokens needed to move the price to the target value) and Δy (number of tokens to receive in return after the swap), one needs to know the type of AMM that DEX uses to price assets. In Appendix A and Appendix B, we have derived equation for Δx and Δy for constant product and stableswap AMMs. Overall, all these parameters can be precisely found and no assumptions need to be made. There is one more parameter that is crucial to take into account—the number of blocks *m* during which attacker will try to manipulate the price. This can not be known upfront but its value affects the cost of an attack significantly. One would suppose that the value of *m* should be small because any deviation from the spot price would be noticed by arbitrageurs and set back to the real value, not allowing an attacker to manipulate for many blocks. However, in pools with infrequent trading activity, the arbitrage opportunity might go unnoticed for longer times. Moreover, there is a chance of multi-block MEV-style attack, where an attacker could cooperate with the miner to mine a few blocks in a row. This style of attack combined with the oracle price manipulation makes the oracle attack cost cheaper. In the examples and simulations below, we assume no MMEV attack and frequent trading activity in a pool.    
**Algorithm 1:** Model to estimate the safety of DEX-based oracle**Input**: Protocol risk parameters, WS, *m*, Δx, Δy, *L***Output**: AC, *C*_**1**_ Calculate minimum deviation from the spot price needed to profit based on protocol-specific parameters._**2**_ Calculate target manipulation price from oracle $TWAP_{m}$.
$$TWAP_{m}=TWAP_{s}+TWAP_{s}\times \epsilon$$_**3**_ Calculate how big should be the manipulation price $p_{m}$ to achieve needed $TWAP_{m}$. For this:
(I)Estimate the oracle window size WS.(II)Decide on number of blocks *m* to manipulate the price.(III)Depending on aggregation method, calculate $p_{m}$ using Equation (Equation 12) or Equation (Equation 14)._**4**_ Calculate attack cost AC based on AMM-specific parameters (liquidity *L*) and $p_{m}$ found above. Find Δx and Δy according to the AMM type, as shown in Appendix A and Appendix B.
AC=Δx−Δy_**5**_ Calculate the minimum capital needed to obtain profit Profit>0 depending on protocol’s risk parameters._**6**_ Estimate the economic feasibility of an attack based on the attack cost AC and collateral *C* values.

### 4.2. Lending Protocols

Lending protocols (also called money markets, credit protocols or protocols for loanable funds) are a market that matches borrowers and lenders—users—who wish to gain interest on their savings, deposit their funds to the lending protocol and then it allows borrowers to lend available assets paying certain interest rate. Detailed explanation on how lending protocols work was provided in the paper [35]. Overall, lending protocols have attracted a lot of interest and become very popular among DeFi community—Ethereum-based lending protocols such as Aave [36], Compound [37], dYdX [38] and MakerDAO [39]. Credit protocols are one of the most popular use cases for AMM-based DEX data to be used as a price feed. Chainlink type of price databases are not always available for relatively new blockchains. In such cases, DEXs, acting as the only option for nascent chains, are used as substitutes for more robust oracle solutions. Considering the large TVL (total value locked in protocol) associated with the popularity of credit protocols and their growing functionality and complexity, it is vital to understand the safe settings of AMM pools that are used as a price information source.

In lending protocols, any user can anonymously borrow funds, but to be able to do so, they first need to provide some collateral asset. To ensure the safety and the solvency of protocol, the Loan-to-Value (LTV) parameter is used—this parameter shows how much a user can borrow relative to their collateral value (all loans in lending protocols are *overcollateralized*. For example, if user deposited 100 USD worth of collateral *C* and LTV parameter is 80%, then they can borrow up to 80% worth of the other asset *B*. More detailed explanations of lending protocols and their risk parameters can be found in [35,36,37,39].

In practice, lending protocols are the most frequent target for oracle manipulation attacks. An attacker tries to artificially increase their collateral value by compromising the oracle price information to be able to borrow more.

We assume a scenario where the attacker artificially increases the value of their collateral to be able to borrow more than their actual collateral value allows. In this case, attacker’s profit can be formulated as follows:(16)Profit=(C×LTV+C×LTV×ϵ)−C

Here, ϵ is the target price manipulation fraction and *C* is the value of collateral—for convenience and normalization purposes, we consider it not as an absolute value but relative to the pool liquidity. This normalization allows us to generalize findings and give parameter recommendations for any pools regardless the size:(17)$$C=\frac{Collateral}{PoolLiquidity}$$

Because the attacker would need to give up their collateral in order to realize their profit from manipulation, we subtract the actual value of their collateral from the profit. From Equation (Equation 16), it is clear that the lower the LTV parameter, the more difficult it is to get the profit from an attack and the higher the ϵ should be. We can derive the value of the target manipulation price from the Equation (Equation 16)—we set the Profit=0 and calculate the ϵ as:(18)$$\epsilon \geq \frac{1}{LTV}-1$$

Figure 1 shows the minimum manipulation target ϵ an attacker needs to achieve for the attack to be profitable given a certain LTV.

Next, after we know the minimum price target needed to make the attack profitable, we can calculate the total cost of an attack using the equations derived in Appendix A and Appendix B.

### 4.3. Attack Scenarios to Lending Protocol Using Constant Product AMM-Based TWAP Oracle

Knowing both the profit and attack cost equations, it is straightforward to simulate various attacks to lending protocols that are using any type of AMM as oracle. We looked at the two most popular types of AMM used in decentralized exchanges—constant product and stableswap—to obtain the full understanding about attacker’s profits and manipulation capital needed.

In this section, we demonstrate how to calculate the cost of an attack under certain conditions using the terms and explanations shown above. Assumptions used in this example are as follows:Loan to Value of the target asset equals 40%.TWAP window equals 30 min.Time without arbitrage equals 1 min.

Note that we are not making any assumptions regarding pool liquidity given how Equation (Equation 16) was defined, which allows us to make calculations, irrespective of the pool liquidity.

Using the attack cost formula and simulating scenarios with varying attacker’s collateral *C*, price target ϵ and pool’s liquidity *L*, we arrive at the following profitability matrix shown in Figure 2, where the space in red indicates a loss (negative profit), the space in blue indicates a positive profit for the attacker and the white area indicates zero-profit scenarios.

From the results shown in Figure 2, we see that the attacker can theoretically reach a profit under almost every combination of events. Moreover, we see that an attack can be profitable by adjusting the requirements to the collateral and the manipulation target ϵ. The lower the ϵ, the higher the collateral the attacker needs to provide for the attack to become profitable and vice versa. These results make us question whether there is such a combination of ϵ and collateral that allows an attacker to obtain profit from an attack with minimum resources?

Figure 3 shows that, effectively, we can retrieve the attack’s minimum cost through a specific combination of ϵ and collateral provided; let us call it an *optimal target*. For the example covered in this section, this point happens at collateral being around five times larger than the pool’s liquidity and manipulation target ϵ being 4.7. Most importantly, the figure below shows that the total capital needed for a profitable attack is 9.3 times the liquidity in the pool used for the AMM.

The optimal target found above and amount of resources needed to reach that point can serve as a reference when deciding on the safety of an oracle.

### 4.4. Attack Scenarios to Lending Protocol Using Stableswap AMM-Based TWAP Oracle

In the previous section, we showed how parameters need to be set for the constant product AMM oracle. In this section, we look at the attack cost and profit when using the stableswap pool as an oracle.

With the attack cost calculated for the stableswap in Appendix A and Appendix B, we can run simulations as in the previous section and produce the profitability matrix. Note that the assumptions used within this section (except for the amplification factor, which is unique to the stableswap AMM) are the same as those used in the previous section. The following figure shows the profitability matrix using a stableswap with an amplification factor *A* of 30.

From Figure 4, it can be seen that the profitability space for an attack is larger in a stableswap AMM than a CPAMM. In other words, manipulating a stableswap-based TWAP is cheaper than a CPAMM-based TWAP oracle using the same assumptions.

Figure 5 shows the minimum cost of performing a profitable attack, indicating a significantly lower point of minimum cost for an attack in a stableswap AMM than in a constant product AMM. Moreover, another difference is the considerably slower growth rate of the attack cost as the manipulation target increases, which makes the total cost of the attack stagnate as the manipulation target increases. In contrast with constant product AMM, the total cost of an attack keeps increasing. For a stableswap AMM, this results in relatively cheap attack opportunities.

As a final note, a stableswap pool can be relatively stable (in terms of price) at a very unbalanced state (in terms of underlying reserves). At the extreme, we could have a situation where the pool is very close to the “knee” of the pricing curve (where the constant sum (linear) part of pricing curve meets the constant product part), and manipulation attacks become increasingly easier to perform given the aggressive nature of the stableswap curve. In other words, we cannot assume that the attack will take place from a 50:50 state or anything closer to that. The more unbalanced the pool at the start of the manipulation attack, the less resources needed to conduct the attack. Therefore, we do not recommend using stableswap pools as an oracle. Please also refer to Appendix B, where various attack scenarios in stableswap pool were shown under different LTV values—it is clear that this type of pool is much cheaper to manipulate comparing with the constant product market. Moreover, as we can see from Figure 5, once the *optimal price target* is reached, attacker does not need significantly more resources to manipulate price higher and to obtain even higher returns from the attack.

## 5. Conclusions and Discussion

DeFi protocols, as well as any other blockchain applications, function in a closed environment, and for their proper performance, a reliable data source (oracle) is needed. Currently, there are two different ways to fetch the data about assets’ prices—either by using trust-based oracles (e.g., Chainlink) or by getting the prices directly from the decentralized exchange. In our research, we focused on understanding the mechanism of the latter option. Understanding the safety of a DEX-based oracle starts from the deep understanding on how DEXs work; nowadays, they function using the automated market-making (AMM) mechanisms and the asset’s price discovery happens along the curve of the AMM cost function. We reviewed the most widely used AMM cost functions and derived the cost of an attack for them. The next step was to look at the various aggregation methods; because using the spot price directly from DEX can lead to cheap price manipulations, most of the DeFi applications aggregate historical spot prices over a certain window size to decrease the chance of an attack. Depending on the method implemented and the window size, the target manipulation price can be higher or lower. We have provided equations to estimate the target attack price based on the aggregation method. We then developed the algorithmic model to estimate the safety of a DEX-based oracle on the example of a lending protocol. A step-by-step algorithm considers protocol-specific, oracle-specific and DEX-specific parameters and provides the logic on how to proceed with deciding on the safety of an oracle. Although we used the lending protocol as an example of a DeFi application using a DEX-based oracle, the model we introduced can be easily generalized to other types of protocols by changing the protocol-specific parameter (LTV in our example).

Incidents that happen in the new field of decentralized finance often lead to the crisis of trust from users and have a large social impact on the entire industry. Despite the crucial role oracles play in decentralized finance, their underlying mechanics are still under-explored and poorly understood which resulted in several protocol exploits [13,14,15]. However, we see the growing interest from both academia and industry practitioners to improve the oracles’ resistance to manipulation attacks—new AMM curves are being introduced [40,41,42], oracle research is growing and more protocols are aware of price manipulation attacks. There is still a lot of work that can be done to achieve the goal of a safe decentralized price oracle in every layer—protocols using oracles can improve their risk management strategies, the AMM cost function can contribute a lot to the safety of oracles, as we saw in Section 4.3 and Section 4.4, where the different pricing curves result in the different costs of attack. Finding the optimal AMM cost function that would minimize the chances of manipulation is not a trivial task, and new pricing curves are being proposed by academia [40,42] and implemented in practice [41]. We hope to see more work performed in this direction. More research can be conducted about information aggregation methods as well—for economic reasons, currently, protocols are using simple statistical methods such as the TWAP. Finding an optimal solution that is less sensitive to the outliers and at the same time has a high price precision and cheap gas cost is still an open question at the moment. Overall, oracles in decentralized finance remain one of the most important and under-researched topics in the field with a huge impact on the entire cryptocurrency system.

## Figures and Tables

**Figure 1 entropy-25-00060-f001:**
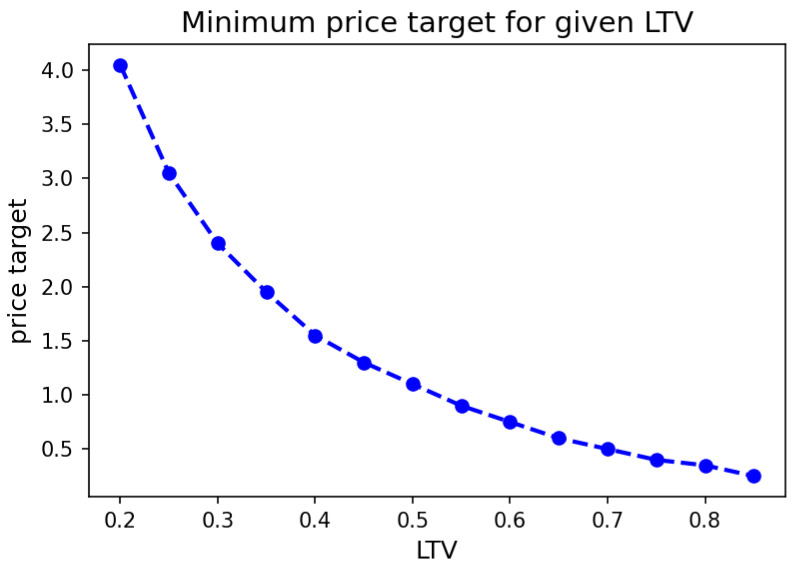
Minimum price target needed for given LTV to obtain the profit from the attack.

**Figure 2 entropy-25-00060-f002:**
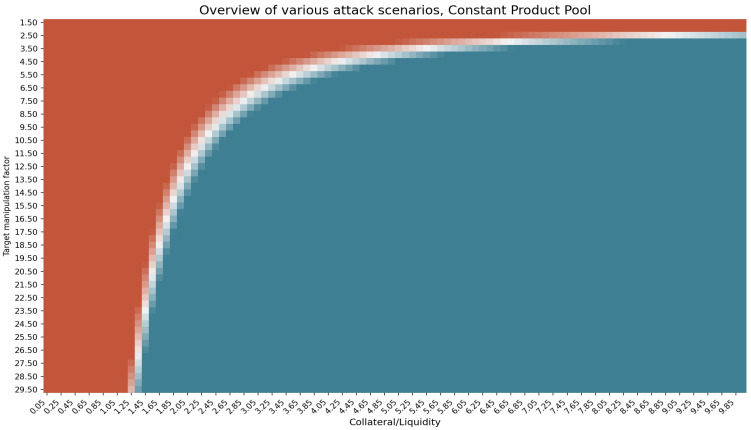
Overview of various attack scenarios in constant product market. The x-axis shows the amount of an attacker’s collateral in terms of liquidity, and the y-axis is the target manipulation price ϵ. The space in red shows the non-profitable attack scenarios (when attack cost exceeds the profit). Blue areas show profitable attacks, while the space in white shows when Profit−AttackCost is close to zero. LTV = 0.4 for all scenarios.

**Figure 3 entropy-25-00060-f003:**
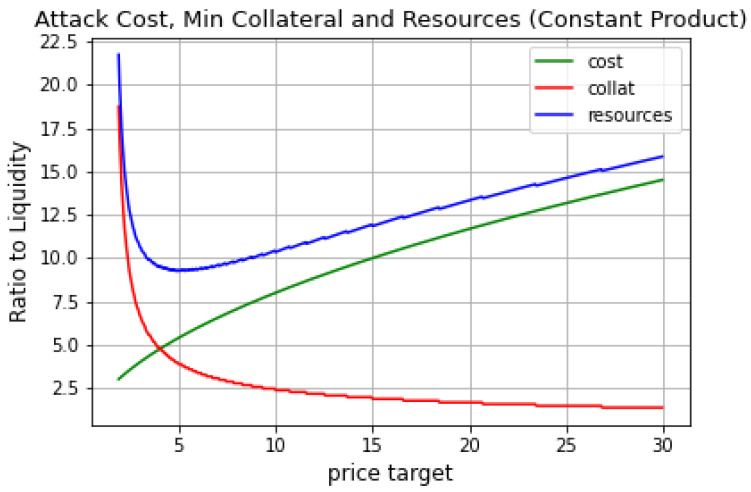
Attack cost, minimum collateral needed and the total resources needed for the profitable attack.

**Figure 4 entropy-25-00060-f004:**
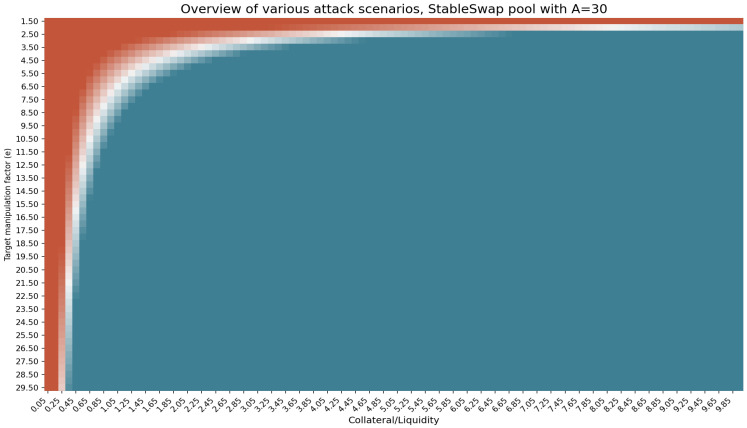
Overview of various attack scenarios in stableswap market. The x-axis shows the amount of attacker’s collateral in terms of liquidity, and y-axis is the target manipulation price ϵ. The space in red shows the non-profitable attack scenarios (when attack cost exceeded the profit). Blue areas show profitable attacks, while the space in white shows when Profit−AttackCost is close to zero. LTV = 0.4 for all scenarios.

**Figure 5 entropy-25-00060-f005:**
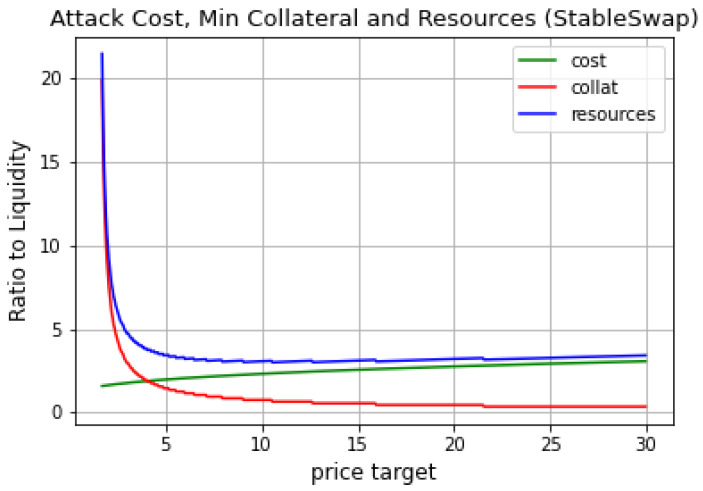
Attack cost, minimum collateral needed and the total resources needed for the profitable attack.

## Data Availability

Not applicable.

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
