# Peer review of "Oracles in Decentralized Finance: Attack Costs, Profits and Mitigation Measures"

_entropy, 2022, doi:10.3390/e25010060_

Round 1

Reviewer 1 Report

1. The main contribution of this paper might be the Algorithm 1. The authors used Lending protocol as an example to design the algorithm and mentioned that the algorithm can be generalized to other types of protocols by changing the protocol-specific parameter. From my point of view, it is very strange to design an algorithm specific to a special case and said that it can be applied to other cases. I think the authors need to proposed a general-purpose algorithm and take a specific protocol as an example to show how to use their algorithm to estimate the safety of DEX-based oracle.

2. The output of Algorithm 1 is AC and C. However, the final step of the algorithm is to estimate the economic feasibility of an attack  based on AC and C. Based on the final step, the appropriate output of the algorithm should be the economic feasibility. Additionally, the authors did not clearly define the economic feasibility. Is it a quantitative or a qualitative measurement? If it is quantitative, how to calculate the value and how to interpret the value? If it is qualitative, how to use AC and C to derive the qualitative value?

3.The authors said in the abstract that "understanding and implementing the logic in the model can help to reduce the chances of price-manipulations attacks". However, It is the authors' responsibility to explain why the chances of price-manipulations attacks can be reduced and how to reduce the chances of price-manipulations attacks through understanding and implementing the logic in the model.

4. Based on the aim of this research, I would recommend to add a section to discuss theoretical, management, social implications, as well as limitations and future avenues in the final section.

Author Response

  1. The main contribution of this paper might be the Algorithm 1. The authors used Lending protocol as an example to design the algorithm and mentioned that the algorithm can be generalized to other types of protocols by changing the protocol-specific parameter. From my point of view, it is very strange to design an algorithm specific to a special case and said that it can be applied to other cases. I think the authors need to proposed a general-purpose algorithm and take a specific protocol as an example to show how to use their algorithm to estimate the safety of DEX-based oracle.

We thank reviewer for suggesting this way of presenting the model. Indeed, it is more logical to present the general algorithm first and then provide an example. In the revised version we first describe the model in Subsection 4.1 and present the algorithmic model as generalized version (Algorithm 1 on page 9). We have moved the description of Lending Protocols to the Subsection 4.2 and provided examples how model can be used when Lending Protocol is using TWAP based oracle to the Subsection 4.3 (Constant Product AMM based TWAP oracle) and Subsection 4.4 (Stableswap AMM based TWAP oracle).

  1. The output of Algorithm 1 is AC and C. However, the final step of the algorithm is to estimate the economic feasibility of an attack  based on AC and C. Based on the final step, the appropriate output of the algorithm should be the economic feasibility. Additionally, the authors did not clearly define the economic feasibility. Is it a quantitative or a qualitative measurement? If it is quantitative, how to calculate the value and how to interpret the value? If it is qualitative, how to use AC and C to derive the qualitative value?

We thank the reviewer for spotting this issue. Indeed, the final goal of the model is to estimate the economic feasibility of an attack and since we did not explain in detail of what is the “economic feasibility” of attack, it might sound odd to put AC and C as final model outputs. We added the explanation of “economic feasibility” in paper at the beginning of Section 4 and explained why we calculate the AC and C as final outputs.

We explained that calculating AC and C as the output gives exact quantitative number while giving the reader/protocol user flexibility to assess the feasibility based on their risk tolerance. However, we still provided an example how economic feasibility can be estimated based on the model’s outputs leaving the reader/protocol user the freedom to adjust according to their own logic.

3.The authors said in the abstract that "understanding and implementing the logic in the model can help to reduce the chances of price-manipulations attacks". However, It is the authors' responsibility to explain why the chances of price-manipulations attacks can be reduced and how to reduce the chances of price-manipulations attacks through understanding and implementing the logic in the model.

We have added more explanations on this in the Introduction Section, paragraph 5 on page 2. Also, additional explanations added in response to the previous comment would also help the reader to understand why and how the chances of price-manipulations attacks can be reduced.

  1. Based on the aim of this research, I would recommend to add a section to discuss theoretical, management, social implications, as well as limitations and future avenues in the final section.

We thank for this raesonable suggestion and we added a section n on page n with a discussion of implications and future directions. We hope this addition would help readers to understand better the importance of the study.

Reviewer 2 Report

The paper presents a well - structured work regarding the performance of AMM - based DEXs used as oracles along with the security risks that this brings.

The paper is well written and the arguments are presented clearly. The authors should be careful for some minor mistakes that affect the reader. For example, the explanation of AMM is first delivered in line 97 while it has been mentioned a lot before.
Also, the security aspect could be further highlighted in the paper along with more suggestion and improvements.

The references are of good quality and updated.

Author Response

Reviewer 2.

1. The paper presents a well - structured work regarding the performance of AMM - based DEXs used as oracles along with the security risks that this brings.

The paper is well written and the arguments are presented clearly. The authors should be careful for some minor mistakes that affect the reader. For example, the explanation of AMM is first delivered in line 97 while it has been mentioned a lot before.

Thanks for the positive feedback! We have corrected minor issues and ex- plained terms before they are presented in the paper.

2. Also, the security aspect could be further highlighted in the paper along with more suggestion and improvements.

The references are of good quality and updated.

We have added more discussions and explanations on oracles security in Introduction and Conclusions part to further highlight the importance of this aspect.
